# Prox1 Suppresses the Proliferation of Breast Cancer Cells via Direct Inhibition of c-Myc Gene Expression

**DOI:** 10.3390/cells12141869

**Published:** 2023-07-17

**Authors:** Artemis Michail, Dimitrios Gkikas, Dimitris Stellas, Valeria Kaltezioti, Panagiotis K. Politis

**Affiliations:** 1Center for Basic Research, Biomedical Research Foundation of the Academy of Athens, 4 Soranou Efesiou Str., 115 27 Athens, Greece; artemisbio@gmail.com (A.M.); dgkikas@bioacademy.gr (D.G.); vkaltezioti@bioacademy.gr (V.K.); 2Department of Biology, University of Patras, 265 04 Patras, Greece; 3Institute of Chemical Biology, National Hellenic Research Foundation, 48 Vassileos Constantinou Ave., 116 35 Athens, Greece; dstellas@eie.gr; 4School of Medicine, European University Cyprus, Nicosia 2404, Cyprus

**Keywords:** metabolism, gene regulation, Warburg effect, PDK1, MPC1

## Abstract

Breast cancer is one of the most lethal malignancies in women worldwide and is characterized by rapid growth and low survival rates, despite advances in tumor biology and therapies. Novel therapeutic approaches require new insights into the molecular mechanisms of malignant transformation and progression. To this end, here, we identified Prox1 as a negative regulator of proliferation and tumor-related metabolism in breast cancer. In particular, we showed that breast tumors from human patients exhibited reduced levels of Prox1 expression, while high expression levels of Prox1 were associated with a favorable prognosis in breast cancer patients. Moreover, we experimentally demonstrated that Prox1 was sufficient to strongly suppress proliferation, migration, and the Warburg effect in human breast cancer cells without inducing apoptosis. Most importantly, over-expression of Prox1 inhibited breast tumor growth in vivo in both heterotopic and orthotopic xenograft mouse models. The anti-tumorigenic effect of Prox1 was mediated by the direct repression of c-Myc transcription and its downstream target genes. Consistently, c-Myc over-expression from an artificial promoter that was not targeted by Prox1 reversed Prox1’s anti-tumor effects. These findings suggest that Prox1 has a tumor suppressive role via direct transcriptional regulation of c-Myc, making it a promising therapeutic gene for breast cancer.

## 1. Introduction

Prox1 is a vertebrate homeobox transcription regulator that plays an important role in embryonic development. Mouse embryos lacking Prox1 die early in embryogenesis at E14.5 [1]. Prox1 is a regulatory factor in lymphatic, liver, pancreas, heart, lens, retina, and nervous system development [1,2,3,4,5,6,7,8,9,10]. Furthermore, Prox1 has been implicated in a large number of human cancers, including blood, pancreatic, liver, colon, lung, and nervous system-related tumors, displaying both suppressing and oncogenic roles in a tissue dependent manner [11,12,13,14,15,16,17,18,19,20,21,22,23]. It has also been previously shown that Prox1 expression is repressed in breast cancer cells due to epigenetic silencing [24]. Consistently, a recently reported Genome-Wide Association Study (GWAS) found a susceptibility locus over Prox1 regulatory sequences that significantly increases the risk for breast cancer in children that received radiotherapy [25]. In addition, an unbiased shRNA screen identified Prox1 as a negative regulator of the proliferation and stemness of mammary stem/progenitor cells [26]. Although these data raise the hypothesis that Prox1 may be negatively associated with tumorigenesis, its role and mechanism of action in breast cancer remain elusive.

On the other hand, abnormal metabolism is a critical hallmark of cancer cells [27,28], including breast cancer. Cancer cells utilize aerobic glycolysis (known as the Warburg effect) to benefit their growth in multiple ways [29,30,31]. The Warburg effect is manifested by a robust increase in glucose uptake and lactate production. It facilitates cancer cell proliferation and migration and enhances tumor growth. It also provides a significant advantage in a low-energy supply environment by contributing intermediate metabolites as a carbon source for anabolic processes [31,32]. Transcription regulators with the ability to promote or facilitate tumorigenesis, such as c-Myc and HIF1a, are able to directly induce the expression of genes encoding for glycolytic enzymes. These regulatory actions constitute critical steps for the onset and progression of tumor cell proliferation, growth, and metastasis. Specifically in breast cancer, the oncogenic transcription factor c-Myc reprograms the cellular metabolism towards aerobic glycolysis by directly transactivating a number of glycolytic genes, including *GLUT1*, *HK2*, *PDK1,* and *LDHA* [33,34,35]. This regulatory axis is a critical inducer of breast tumorigenesis, and cancer therapeutic approaches are focused on drugs targeting the c-Myc and Warburg effect [36,37,38,39,40]. Although the molecular pathway via which c-Myc promotes the metabolic hallmarks of cancer cells has been extensively studied, the tumor suppressive mechanisms that directly prevent this oncogenic axis are not known. Unraveling the role of these mechanisms could significantly contribute to future therapeutic approaches.

Here, we showed that Prox1 exhibits a tumor-inhibiting function in breast cancer by directly repressing the c-Myc gene expression at the transcriptional level. Via this action, Prox1 strongly suppresses human breast cancer cell growth, Warburg effect and invasion, in vitro and in vivo, in a non-apoptotic way. It also promotes a gene expression program characterized by down-regulation of c-Myc-controlled glycolytic genes that mediate aerobic glycolysis repression. Collectively, these findings indicate a tumor-suppressive role of Prox1 through the direct suppression of c-Myc-induced metabolic reprogramming of cancer cells.

## 2. Methods

### 2.1. Ethics Statement

The committee of local ethics approved our research protocol (Athens Prefecture Veterinarian Service: 5523/16-10-2018), which was then carried out at the Center for Experimental Surgery (animal facilities) of the BRFAA. The treatment of all animals adhered strictly to the guidelines of good animal practice set by the appropriate animal welfare organizations in Europe and Greece.

### 2.2. Reagents

The reagents that we used for cell culture were: FBS (Biosera, Cholet, France), streptomycin/penicillin (Invitrogen, Waltham, MA, USA), horse serum (Biosera, Cholet, France), Cell Media (Biosera, Cholet, France). For all other assays we used: Lipofectamine 2000 (Invitrogen, 11668027, Waltham, MA, USA), Crystal Violet (Sigma, Burlington, MA, USA), TRI reagent solution (AM9738, Ambion/RNA, Life Technologies, Carlsbad, CA, USA), RQ1 DNase (Promega, Madison, WI, USA), Superscript First-Strand Synthesis System kit (Invitrogen, Waltham, MA, USA), Platinum SYBR Green qPCR supermix-UDG kit (Invitrogen #11733–046, Waltham, MA, USA), nitrocellulose membranes (Amersham, Slough, Buckinghamshire, UK), BSA (AppliChem, A1391, Darmstadt, Germany), FACS buffer (7), Nucleospin RNA XS kit, (Macherey Nagel, Hoerdt Cedex, France), poly-L-lysine (Sigma, Burlington, MA, USA), buprenorphine (Abbott Laboratories, Illinois, United States), autoclips (Stoelting, Wood Dale, Chicago, IL, USA), tissue adhesive (Histoacryl, B.Braun, Melsungen, Germany), meloxicam (Pfizer, New York City, NY, USA), contour next blood glucose monitoring system (meter, test strips and control solution), the Accutrend BM-lactate test strips with the Accutrend Plus meter from Roche (Art. No 128633, Basel, Switzerland), protein-A-G magnetic beads (Invitrogen, Dynabeads, Waltham, MA, USA), and the Qiagen PCR purification minelute kit. The primary antibodies we used were rabbit anti-c-Myc (Abcam, ab32072, Waltham, Boston), mouse anti-HK2 (Santa Cruz, sc-130358, Dallas, TX, USA), rabbit anti PDK1 (proteintech 10026-1-AP, Rosemont, IL, USA), mouse anti-MPC1 (proteintech 66272-1-Ig, Rosemont, IL, USA ), rabbit anti-phospho-Histone 3 (Abcam, Ab5176, Waltham, Boston), rat anti-BrdU (Abcam, 6326, Waltham, Boston), rabbit anti-cleaved caspase 3 (Cell Signaling, 9661, Danvers, MA, USA), mouse anti-Ki67 (Cell Signaling, 9449, Danvers, MA, USA), rabbit anti-Prox1 (ReliaTech, 102-PA32, Los Angeles, CA, USA), rabbit anti-VEGF (R&D Systems, AF-493, northeast Minneapolis), rabbit anti-OCCLUDIN (Invitrogen, 71-1500, Waltham, MA, USA), rabbit anti-CD31 (Abcam, ab-28364, Waltham, Boston), and mouse anti-beta actin (Sigma, A5441, Burlington, MA, USA). The secondary antibodies we used were: donkey anti-mouse 488 (Alexa Fluor, Thermo Fisher, Waltham, MA, USA), goat anti-rabbit IgG (Sigma, A6154, Burlington, MA, USA), rabbit anti-mouse IgG (Sigma, A9044, Burlington, MA, USA), donkey anti-mouse 568 (Alexa Fluor, Thermo Fisher, Waltham, MA, USA), goat anti-rabbit 488 (Alexa Fluor, Thermo Fisher, Waltham, MA, USA, donkey anti-rabbit 647 (Alexa Fluor, Thermo Fisher, Waltham, MA, USA), donkey anti-rat 555 (Alexa Fluor, Thermo Fisher, Waltham, MA, USA), donkey anti-rabbit 568 (Alexa Fluor, Thermo Fisher, Waltham, MA, USA), and donkey anti-chicken 488 (Alexa Fluor, Thermo Fisher, Waltham, MA, USA).

### 2.3. Biological Resources

In the current study, we used human breast cancer cell lines provided by the American Type Culture Collection (ATCC) company. MDA-MB-231 cell line possesses epithelial cell morphology (specifically adenocarcinoma) and it has its origin in mammary gland tissue (cat. Number: HTB-26). MCF7 cell line possesses epithelial cell morphology (specifically adenocarcinoma) and it has its origin in mammary gland tissue (cat. Number: HTB-22). MCF7 cell line possesses epithelial cell morphology (specifically ductal carcinoma) and it has its origin in mammary gland tissue (cat. Number: HTB-133). We also used a non-cancerous breast cancer cell line, MCF10A, which possesses epithelial cell morphology (specifically fibrocystic disease) and it has its origin in mammary gland tissue (cat. Number: CRL-10317). The human breast cancer cell lines are reported by their official names as listed in ExPasy Research Database. The human breast cancer cell lines (MDA-MB-231, MCF7, T47D) were cultured in the recommended medium supplemented with 10% FBS (Biosera, Cholet, France), 1% streptomycin/penicillin (Invitrogen, Waltham, MA, USA), and incubated in 37 °C humidified incubator with 5% CO_2_. The human breast cell line MCF10A was cultured in the recommended medium supplemented with 5% Horse Serum (Biosera, Cholet, France), 1% streptomycin/penicillin (Invitrogen, Waltham, MA, USA), 20 ng/mL EGF (Invitrogen, Waltham, MA, USA), 10 μg/mL insulin (Invitrogen, Waltham, MA, USA), 100 ng/mL cholera toxin (Sigma Aldrich, Burlington, MA, USA), and incubated in 37 °C humidified incubator with 5% CO_2_. All human breast cancer cells lines used in this study have been authenticated by their differential expression of ER, PR, and HER2 receptors and are mycoplasma-free.

The mouse breast cancer primary cells (mmtv-Myc), derived from MMTV-Myc murine breast tumors [41], were cultured in DMEM-High Glucose medium supplemented with 10% FBS (Biosera, Cholet, France), 1% streptomycin/penicillin (Invitrogen, Waltham, MA, USA), and incubated in 37 °C humidified incubator with 5% CO_2_.

### 2.4. Transfection and Infection Methods

All cell lines were transfected with Lipofectamine 2000 (Invitrogen, 11668027, Waltham, MA, USA) according to the manufacturer’s protocol.

For adenoviral infection of MDA-MB-231 cells and MCF7 cells, 6 × 10^5^ cells were incubated for 18 h with 50 μL of crude adenoviral medium and harvested after 48 h. The recombinant control over-expressing GFP (Ad-GFP) and the recombinant experimental over-expressing Prox1 (Ad-Prox1) adenoviruses were constructed using the pAd/PL-DEST Gateway vector (ViraPower Adenoviral Expression System, Invitrogen Life Technologies, Carlsbad, CA, USA) following the instructions provided by the manufacturer. We also previously used and reported these adenoviral systems for the infection of other mammalian cells [42,43]. In summary, the cDNAs encoding wild-type Prox1 and eGFP were inserted into a modified version of the pENTR.GD entry vector and were then transferred into the Destination vector. The recombinant adenoviral particles were generated in the HEK-293A cell line, and the virus titers were determined using the plaque assay method [44].

### 2.5. Transwell Chamber Assay

In the transwell chamber assay, we used cell inserts with a membrane pore size of 8.0 μm (Corning 3422). Breast cancer cells were serum-deprived for 12 h and then seeded onto the upper surface of the chamber in serum-free medium, with a cell density of 1 × 105 cells per well. The lower chamber contained 20% FBS (fetal bovine serum) and the entire chamber was incubated at 37 °C for 24 h. Following incubation, we removed the non-motile cells of the upper surface of the filter, while we fixed with MeOH the cells on the lower chamber, stained them with Crystal Violet (Sigma, Burlington, MA, USA), and counted them. Cell counting was performed in triplicate inserts by examining 5 random fields using bright-field microscopy. The data were then analyzed using Image J software, Version 1.53 (Fiji).

### 2.6. RNA Extraction and Real Time RT-qPCR Analysis

Total RNA was extracted using TRI reagent solution (AM9738, Ambion/RNA, Life Technologies, Carlsbad, CA, USA) following the manufacturer’s instructions, as we previously reported [5,42]. The extracted RNA was then treated with RQ1 DNase (Promega, Madison, WI, USA) for any residual DNA removal. The concentration and purity of the RNA were determined using a Nanodrop 2000c (Thermo), and 1.5 μg of the RNA was used for cDNA synthesis. The cDNA synthesis was carried out using the SuperScript First-Strand Synthesis System (Invitrogen, Waltham, MA, USA) along with random hexamer primers. Quantitative Real-time RT-PCR analysis was performed using a LightCycler 96 Instrument (Roche, Basel, Switzerland) [42,43]. The Platinum SYBR Green qPCR supermix-UDG kit (Invitrogen #11733-046, Waltham, MA, USA) was used to perform real-time PCR in a 20 μL reaction, following the manufacturer’s recommendations. The specific primers used for performing real time RT-qPCR are noted in Table 1. The normalization of the measured values was performed by using hRPL13 and hPPIA mRNA levels as internal references.

### 2.7. Western Blot Analysis

Cells were treated with RIPA lysis buffer to extract total protein and the resulting homogenates were centrifuged at 17,000× *g* for 15 min at 4 °C. Only the supernatants were collected, and for the protein concentration measurement, we used the Bradford protein assay (Bio-Rad protein assay). For each sample, 35 μg of protein was loaded onto SDS-PAGE gels and subsequently transferred to nitrocellulose membranes (Amersham, Slough, Buckinghamshire, UK) using a semi-dry transfer system (Bio-Rad). The membranes were then blocked with 5% BSA (Applichem, A1391, Darmstadt, Germany) dissolved in Tris-buffered saline (1×) containing 0.1% Tween-20 for 1 h at RT. Following the blocking time, the membranes were incubated with primary antibodies overnight at 4 °C, and then with secondary antibodies for 1.5 h at RT. The primary antibodies that were utilized for the Western blot experiments are described below: rabbit anti-c-Myc (Abcam, ab32072, Waltham, Boston ) (1:500 dilution), mouse anti-HK2 (Santa Cruz, sc-130358, Dallas, TX, USA) (1:200 dilution), rabbit anti PDK1 (proteintech 10026-1-AP, Rosemont, IL, USA) (1:1000 dilution), mouse anti-MPC1 (proteintech 66272-1-Ig, Rosemont, USA) (1:1000 dilution), and mouse anti-beta actin (Sigma, A5441, Burlington, MA, USA) (1:20.000 dilution). The secondary antibodies were rabbit anti-mouse IgG (Sigma, A9044, Burlington, MA, USA) (1:20.000 dilution, Burlington, MA, USA) and goat anti-rabbit IgG (Sigma, A6154, Burlington, MA, USA) (1:10.000 dilution).

### 2.8. Fluorescence-Activated Cell Sorting (FACS) Analysis

Cultures of MCF7 human breast cancer cells were transfected using IRES-GFP or Prox1-IRES-GFP constructs. Then, the cells were further cultured for an extra 2 days after transfection. On the third day, cells were washed with DMEM with 1 gr/L glucose, resuspended in FACS buffer [33], 0.5% FBS, and 1 mM EDTA; pH = 8], and they were filtered using a 50 μm filter (Falkon). Transgene expressing cells were sorted on BD FACS Aria TM ll (BD biosciences). The process of extracting RNA from isolated cells was conducted using Nucleospin RNA XS (Macherey Nagel, Hoerdt Cedex France) in accordance with the manufacturer’s instructions.

### 2.9. Immunostainings

As far as immunostaining experiments are concerned, cell lines were cultured in 24-well plates onto poly-L-lysine (Sigma, Burlington, MA, USA) coated cover slips. Following transfection or adenoviral infection method, the cells were fixed on the cover slips with 4% paraformaldehyde (PFA) and blocked with 5% FBS dissolved in phosphate-buffered saline (PBS) (1×) containing 0.2% Triton X-100 for 60 min at RT. The next step was incubation with primary antibodies at 4 °C overnight, followed by secondary antibodies incubation for 1 h at RT. The last step was incubation with DAPI diluted in 1 × PBS for 10 min at RT followed by mounting with MOWIOL, as previously described [5]. As far as in vivo allotransplantation experiments are concerned, tumor samples were dehydrated in alcohol after being fixed in 10% formalin solution for 24 h. The tumors were then cleared with xylene, embedded in paraffin, sectioned to a thickness of 10 μm, and collected on poly-D-lysine-coated slides. For immunostaining, the paraffin section was deparaffinized/rehydrated, followed by blocking and antibody incubation as described above. The primary antibodies used in the immunofluorescence experiments described in the present study were: rabbit anti-Prox1 (ReliaTech, 102-PA32, Los Angeles, CA, USA) (1:200 dilution), rabbit anti-cMyc (Abcam, ab32072, Waltham, Boston) (1:100 dilution), rabbit anti-VEGF (R&D Systems, AF-493, northeast Minneapolis) (1:100 dilution), rabbit anti-OCCLUDIN (Invitrogen, 71-1500, Waltham, MA, USA) (1:200 dilution), rabbit anti-CD31 (Abcam, ab-28364, Waltham, Boston), (1:500 dilution), rat anti-BrdU (Abcam, 6326, Waltham, Boston) (1:400 dilution), rabbit anti-cleaved caspase 3 (Cell Signaling, 9661, Danvers, MA, USA) (1:800 dilution), rabbit anti-phospho-Histone 3 (Abcam, Ab5176, Waltham, Boston) (1:600 dilution), and mouse anti-Ki67 (Cell Signaling, 9449, Danvers, MA, USA) (1:1000 dilution). The secondary antibodies that were used to develop the immunofluorescence signal were as follows: donkey anti-mouse 568 (Alexa Fluor, Thermo Fisher, Waltham, MA, USA), donkey anti-rabbit 647 (Alexa Fluor, Thermo Fisher, Waltham, MA, USA), goat anti-rabbit 488 (Alexa Fluor, Thermo Fisher, Waltham, MA, USA), donkey anti-chicken 488 (Alexa Fluor, Thermo Fisher, Waltham, MA, USA), donkey anti-rat 555 (Alexa Fluor, Thermo Fisher, Waltham, MA, USA), donkey anti-rabbit 568 (Alexa Fluor, Thermo Fisher, Waltham, MA, USA), and donkey anti-mouse 488 (Alexa Fluor, Thermo Fisher, Waltham, MA, USA ).

### 2.10. Allotransplantation Experiments (Xenografts)

#### 2.10.1. Orthotopic Xenografts for Breast Cancer

The tumor in vivo orthotopic model was established using 6-week-old female NOD-SCID mice (NOD SCID gamma mouse) from the animal facilities of the Center for Experimental Surgery of the BRFAA. NOD-SCID mice lack mature T cells, B cells, and natural killer (NK) cells and are also deficient in multiple cytokine signaling pathways. Mice were kept in individually ventilated cages under aseptic conditions in a climate (24 °C) and light-controlled setting (12 h each day), having unlimited amounts of autoclaved food and water. For the surgical procedure, mice were anesthetized by a steady flow of oxygen and isoflurane gas (4%) and their body temperature was kept steady at 37 °C by a heating pad. Furthermore, an analgesic mixture of buprenorphine (Abbott Laboratories, Chicago, IL, USA) (0.1 mg/kg) and meloxicam (Pfizer, New York City, NY, USA) (2 mg/kg) was provided to the animals. Following the analgesia, a midline incision through the skin and fascia was made using a sterile scalpel, and then, the fat-pad of both inguinal mammary glands was slightly elevated for the injection of equal numbers of MDA-MB-231 cells (over-expressing Prox1 or GFP) both into the left and right fat-pads, in order to track the growth of both tumor types under the exact same conditions. The last step of the procedure was to close the incision using tissue adhesive (Histoacryl, B.Braun, Melsungen, Germany) and the skin using auto clips (Stoelting Europe). We then monitored and measured the tumor development using sliding calipers every 3 days. In order to calculate the volume of tumors, we used the formula: volume (mm^3^) = (width^2^ × length)/2. Following the mice sacrifice, we dissected and weighted the tumors. Tumors were then isolated and further proceeded for immunostainings, RNA and protein extraction. For the in vivo orthotopic breast cancer mouse model, we used 10 female NOD-SCID mice (n = 10).

#### 2.10.2. Heterotopic Xenografts for Breast Cancer

The tumor in vivo model was established using 4-week-old female NOD-SCID mice (NOD SCID gamma mouse) from the animal facilities of the Center for Experimental Surgery of the BRFAA. NOD-SCID mice lack mature T cells, B cells, and natural killer (NK) cells and are also deficient in multiple cytokine signaling pathways. Mice were kept in individually ventilated cages under aseptic conditions in a climate (24 °C) and light-controlled setting (12 h each day), having unlimited amounts of autoclaved food and water. For the heterotopic experiments, we conducted injections in mice subcutaneously at one dorsal site with 1 × 10^6^ respective MDA-MB-231 cells (over-expressing Prox1 or GFP) in 100 μL PBS. We then monitored and measured the tumor development using sliding calipers every 2 days. For the tumor volume calculation, we used the formula: volume (mm^3^) = (width^2^ × length)/2, as previously mentioned [12,43]. When mice were sacrificed, tumors were dissected and weighed. For the growth test, mice were sacrificed; tumors were isolated and further proceeded for immunostainings, RNA and protein extraction. For the in vivo heterotopic breast cancer mouse model, we used 5 NOD-SCID mice for each condition (GFP and Prox1), and the experimental protocol was performed for two times (n = 20).

### 2.11. Wound Healing Assay

Following the transfection, cells were trypsinized and seeded into 6-well plates to enable the formation of a full monolayer. A 10l pipette tip was used to properly introduce wounds throughout the cell monolayer once they became confluent. 1 × PBS washes were used to remove the dead cells. Then, serum-free media was used in the cell culture for 24 h. Using Image J software, Version 1.53 (Fiji), migration distance was estimated and evaluated after images were collected at 24 and 48 h, respectively.

### 2.12. Glucose and Lactate Assays

After transfection, cells were trypsinized and seeded into 6-well plates to allow the cells to form a complete monolayer. Glucose production was measured by the Contour next blood glucose monitoring system (meter, test strips and control solution). The Contour next system is intended for the quantitative measurement of glucose (from 0.6 mmol/L to 33.3 mmol/L) using approximately 10 μL of the supernatant (1 drop). Lactate production was measured by the Accutrend BM-lactate test strips with the Accutrend Plus meter from Roche (Art. No 128633, Basel, Switzerland), using approximately 10 μL of the supernatant (1 drop). The measurement took place at 24 h, 48 h, and 72 h without changing the medium.

### 2.13. Cross-Linking Chromatin Immunoprecipitation

To analyze the molecular interactions of Prox1 in breast cancer, ChIP experiments were carried out using 10^7^ MDA-MB-231 cells, over-expressing Prox1. The cells were seeded to form a complete monolayer. The next step was to add 0.75% formaldehyde to the media in order to cross-link proteins to DNA, and then, we rotated gently for 10 min at RT. Following the rotation, we added glycine (to stop the cross linking) to final concentration 125 mM and shook at RT for 5 min. After the obtainment of cell nuclei, we lysed them using sonicator to an average length of 250–800 bp in 1% SDS-containing buffer (VCX: 30% amplitude, 1s intervals, 9 min total time, tube on ice). The soluble chromatin was pre-cleared using agarose beads, BSA, and t-RNA, and then, 50 μg of sheared DNA was used with 10 μg of antibody per IP reaction. Following the IP reaction, the complexes of chromatin–antibody were formed at 4 °C by antibody to Prox1 (rabbit anti-Prox1, ReliaTech, 102-PA32, Los Angeles, CA, USA) or control IgG antibody overnight. Using protein-A-G magnetic beads (Invitrogen, Dynabeads, Waltham, MA, USA), the amount of antibody that bounded chromatin was retained. The next step of the protocol was 6 h of reverse cross-linking at 65 °C, digestion with RNAse A (10 mg/mL) and proteinase K (20 mg/mL), and then, DNA purification by the Qiagen PCR purification minelute kit. Quantitive PCR was used for detection and analysis of ChIP precipitates. The Platinum SYBR Green qPCR supermix-UDG kit (Invitrogen #11733-046, Waltham, MA, USA) was used to perform real-time PCR in accordance with the manufacturer’s instructions in a 20 μL qPCR reaction. In each example, the input sample’s data (Ct values) were used for normalization using the percent of input (% IP) approach, and the results were displayed as the fold change in relation to the control anti-IgG IPs, as previously described [12,43]. At least three more ChIP assays were performed for each interaction (Table 2).

### 2.14. Statistical Analysis and Experimental Design

Each subsection of the “materials and methods” details each experimental design. Using IBM SPSS Statistics for Windows, Version 20.0, the Shapiro–Wilk normality test was used to confirm that the data had a normal distribution. All tests were carried out independently three to four times in order to ensure the repeatability of results. For statistical analysis, the two-tailed Student’s *t*-test was used to assess all measurements and experimental values from independent experiments. Every result is shown as mean ± SD. Each Figure legend includes a description of the precise *p* values. Statistics consider *p* values under 0.05 to be significant. Graph Pad 8, Microsoft Excel 2013, and Image J software were used for all analyses.

### 2.15. Web Sites/Data Base Referencing

The Web Sites that we used for metanalysis of expression data from breast cancer patients were: Oncomine (www.oncomine.org, accessed on 15 December 2020), TCGA (The Cancer Genome Atlas, (www.cancergenome.nih.gov, accessed on 1 October 2021), Kaplan–Meier Plotter (https://kmplot.com/analysis, accessed on 1 October 2021), TNM plotter tool (https://tnmplot.com/analysis, accessed on 1 October 2021) and Biorender (https://biorender.com, accessed on 10 May 2022).

## 3. Results

### 3.1. Prox1 Is Significantly Reduced in Breast Cancer and Its Expression Is Correlated to Favorable Prognosis

To initially address the role of Prox1 in breast cancer, we examined the publicly available data from databases such as Oncomine (www.oncomine.org, accessed on 15 December 2020) and TCGA (The Cancer Genome Atlas, www.cancergenome.nih.gov, accessed on 1 October 2021) for clinical associations between Prox1 expression and breast cancer progression. Thus, clinical data from these databases suggest a correlation between low expression levels of Prox1 and breast cancer progression. By using the Oncomine database to analyze TCGA data from breast cancer patients, we observed a strong reduction in *Prox1* gene expression in various subtypes of invasive breast carcinoma as compared to healthy tissue (Figure 1A). Also, using the TNM plotter web tool to examine a larger number of datasets [43], we showed a similar reduction in *Prox1* expression in breast tumor samples (Figure 1Β). Similarly, examination of the data form cBioPortal database further confirmed the reduction in Prox1 expression in all different subtypes of breast tumors as compared to healthy tissue (Appendix A). In agreement, survival analysis of patients using the Kaplan–Meier plotter web tool [44] associated high levels of Prox1 expression in breast tumors with increased survival rates (Figure 1C, and Appendix A). Consistent with these observations, Prox1 expression was much higher in the non-cancerous MCF10A breast cell line as compared to the breast cancer cell lines MDA-MB-231, MCF7, and T47D (Appendix A). Taken together, these data support a hypothesis about the tumor-suppressing role of Prox1 in breast cancer progression.

### 3.2. Prox1 Inhibits Breast Cancer Cell Proliferation without inducing Cell Death

To determine whether the observed correlation between Prox1 expression and favorable prognosis in patients has a functional significance, we analyzed proliferation and apoptosis in human breast cancer cell lines as a model system. In this regard, we first used a lipofection-based strategy and the Prox1-IRES-GFP plasmid vector to perform the over-expression experiments. We previously used and validated the ability of this vector to over-express Prox1 [3,5,45]. We also confirmed it in this study (Appendix A). We then examined the effect of Prox1 over-expression on proliferation of the human MDA-MB-231 cells (Figure 2A–F). We focused our proliferation analysis only on the transfected cells, which we were able to follow due to the GFP signal (denoting either the Prox1-IRES-GFP or IRES-GFP transfected cells). Specifically, we tested the ability of Prox1 to affect the numbers of BrdU+ breast cancer cells in typical BrdU incorporation experiments following a 2 h pulse (before to fixation). We used a 2 h pulse for the BrdU experiments to specifically mark the breast cancer cells that pass through the S-phase of the cell cycle (DNA replication). These experiments revealed a significant decrease in the numbers of BrdU+ cells following Prox1 over-expression (Figure 2A,B), indicating a decrease in the proliferation rates of these cells. (Figure 2A,B). In addition, immunostainings with the phosphorylated-histone H3 (pH3) marker showed a strong reduction in Prox1-IRES-GFP transfected MDA-MB-231 cells undergoing mitosis as compared to IRES-GFP transfected cells (Figure 2D,E). We also measured the proliferation rate in the non-transfected cells from the same specimens (same coverslips) to rule out culture artifacts or non-cell autonomous effects (Figure 2A,C,D,F). In all cases, the non-transfected cells in Prox1 experiments exhibited proliferation rates similar to GFP experiments, confirming that the Prox1-mediated defect in proliferation was only due to the transfected cells. Similar results were obtained using MCF7 cell line (Figure 2G–J).

Next, we wanted to confirm the Prox1-mediated anti-proliferative effect with another over-expression approach. To this end, we utilized an adenoviral-based Prox1 over-expression system and confirmed that this system was capable to over-express Prox1 (Appendix A). Moreover, this system gave us the opportunity to significantly increase the Prox1 over-expression efficiency in the cell culture (the percentages of transduced cells were close to 90%) (Appendix A) and, therefore, perform RNA, protein, or phenotypic analyses without the need to follow only the transfected cells. The adenoviral-mediated over-expression of Prox1 was sufficient to strongly reduce the numbers of proliferating Ki-67+ cells in both MDA-MB-231 and MCF7 cell lines in a manner similar to the lipofection-based approach (Appendix A).

Furthermore, the decreased proliferation of Prox1 over-expressing cells was not accompanied by increased apoptotic rates. Comparing the over-expressing Prox1 cells to the GFP controls, the immunostainings for cleaved caspase 3 and DAPI nuclear staining for apoptotic nuclei revealed no evidence of induced apoptosis. This finding was observed in both cell lines used here (Appendix A). Taken together, these results suggest that Prox1 significantly inhibited the proliferation of breast cancer cells without concurrently promoting cell death.

### 3.3. Prox1 Over-Expression Represses Breast Cancer Cells’ Capacity for Migration and Invasion

Apart from their ability to grow and divide at a rapid rate, another hallmark of cancer cells is their enhanced capacity to migrate and invade other tissues [46]. To evaluate the role of Prox1 in migration and invasion of breast cancer cells, we performed wound healing and Transwell assays. In agreement with a tumor suppressing function, over-expression of Prox1 in both MDA-MB-231 and MCF7 cells significantly inhibited cell migration and invasion (Figure 3).

### 3.4. Prox1 Negatively Affects Glucose Uptake and Lactate Secretion by Breast Cancer Cells

Abnormal metabolism is another critical hallmark of cancer cells [27,28,47]. In particular, cancer cells utilize aerobic glycolysis (known as the Warburg effect) to benefit their growth [29,30,31]. Thus, cancer cells depend on glycolysis and lactate production for ATP generation, growth, and proliferation, even in the presence of sufficient oxygen levels [47]. To further investigate the possible involvement of Prox1 in the metabolic properties of breast cancer cells, we measured the glucose uptake and lactate secretion of Prox1 over-expressing breast cancer cells (Figure 4A). Interestingly, Prox1 over-expression resulted in reduced glucose uptake and lactate secretion in MDA-MB-231 cells (Figure 4B,C). We also confirmed these observations in the MCF7 cell line (Figure 4D,E). These data strengthen the notion that Prox1 could have a negative impact on breast tumor initiation and progression by affecting cancer cell proliferation and metabolism and not by initiating a strong apoptotic cascade.

### 3.5. Prox1 Inhibits Expression of Genes That Promote the Warburg Effect in Breast Cancer Cells

To further investigate the molecular mechanism that underlies the observed anti-proliferative effect of Prox1, we over-expressed Prox1 in the human cell line MCF7 and we then examined the expression of genes involved in breast cancer hallmarks, including proliferation, metastasis, and metabolism. Prox1 over-expression specifically represses c-Myc gene expression (Figure 5A) as well as a number of c-Myc downstream target genes encoding for enzymes in the glycolytic pathway such as *GLUT1*, *HK2*, and *ENO2* (Figure 5A). In the same way, Prox1 inhibits the expression of another downstream target of c-Myc, the *PDK1* gene, which encodes for Pyruvate Dehydrogenase Kinase 1. The function of this kinase is to phosphorylate and inactivate the pyruvate dehydrogenase enzyme (PDH), which catalyzes the conversion of pyruvate into acetyl-CoA and, consequently, its entrance into the Krebs cycle. PDK1 acts to direct the pyruvate toward lactate production at the expense of acetyl-CoA. Therefore, our data suggest that Prox1 inhibited aerobic glycolysis and lactate production. Consistently, Prox1 over-expression induced the expression of genes that inhibit glycolysis or promote the entrance of pyruvate into the Krebs cycle, including the TIGAR (Tp53-Inducible Glycolysis and Apoptosis Regulator) gene and the carrier of pyruvate into mitochondria MPC1 (Mitochondrial Pyruvate Carrier 1), respectively (Figure 5A). Of note, the deregulation of PDK1, TIGAR, and MPC1 was previously strongly correlated with tumor development [48,49,50,51]. Moreover, Prox1 was able to enhance the expression of OXPHOS-related genes such as NDUFA7 (Figure 5A). We also confirmed these effects on metabolism-related genes in another breast cancer cell line, MDA-MB-231 (Appendix A). Additionally, we showed that Prox1 over-expression has the ability to modulate the expression of HK2, PDK1, and MPC1 at the protein level (Appendix A). Collectively, these observations indicate that Prox1 inhibits genes that promote glucose uptake and fermentation of glucose to lactate (the Warburg effect). On the other hand, Prox1 induces the expression of genes that constrain the Warburg effect. These Prox1-mediated effects were observed both on a cell line positive for estrogen and progesterone receptors (MCF7), representing a non-highly metastatic cancer model, and in a triple-negative cell line for all three hormone receptors (MDA-MB-231), constituting a much more aggressive model system [52,53,54].

### 3.6. Prox1 Directly Suppresses c-Myc Gene Expression

c-Myc is the upstream positive regulator of the Prox1-inhibited genes (*GLUT1*, *HK2*, and *PDK1*), identified here, as well as the Warburg effect in cancer cells [55]. Moreover, our mRNA expression analysis in breast cancer cells suggests that Prox1 represses c-Myc expression (Figure 5A). In agreement with this hypothesis, the comparison of endogenous levels of mRNA expression between cancerous (MDA-MB-231, MCF7, and T47D) and non-cancerous (MCF10A) human breast cell lines showed that Prox1 and c-Myc exhibited opposite expression patterns (Appendix A). To further investigate this hypothesis, we examined the ability of Prox1 to reduce the expression of c-Myc at the protein level. Consistently, Prox1 suppressed c-Myc protein and mRNA expression in both MDA-MB-231 and MCF7 cells (Figure 6A–C,F–H). In addition, immunostainings with the c-Myc marker showed a strong reduction in Prox1-positive MDA-MB-231 and MCF7 cells as compared to GFP-positive cells (Figure 6D,E,I,J). To test whether this regulatory action of Prox1 on the c-Myc gene is direct, we performed a series of chromatin immunoprecipitation (ChIP) experiments. First, by using bioinformatic tools, we identified a number of potential binding sites for Prox1 over the human c-Myc gene (Figure 6K). Second, by using ChIP assays in the Prox1 over-expressing MDA-MB-231 cells, we showed that Prox1 is directly recruited at three of these loci over the c-Myc gene (Locus B, Locus D, and Locus F in Figure 6L) near the transcription start site, into the second intron and 3′ end of the gene, respectively. These data indicate that Prox1-mediated transcriptional inhibition of the c-Myc gene occurred via a direct interaction with these specific regulatory sequences (Figure 6M). These interactions may also mediate the tumor-inhibiting function of Prox1 in breast cancer cells.

### 3.7. c-Myc Over-Expression Rescues the Anti-Proliferative Effect of Prox1 on Breast Tumor Cells

To further investigate the c-Myc/Prox1 interaction, we assessed whether c-Myc over-expression from an artificial promoter, lacking the Prox1 binding sites of the endogenous c-Myc gene, could rescue the anti-proliferative effect of Prox1 on breast cancer cells. To achieve this, we co-expressed c-Myc and Prox1 in MCF7 cancer cells and compared them to cells over-expressing only Prox1 or GFP. Noticeably, c-Myc complementation was sufficient to fully restore the ability of breast cancer cells to proliferate (Figure 7A,B). These data indicate that there was a total rescue of Prox1’s anti-proliferative effect on breast cancer cells upon c-Myc complementation. In addition, we isolated primary tumor cells from a mouse genetic model for breast cancer. In this model, breast tumors were induced by the expression of c-Myc from an mmtv promoter (mmtv-Myc) (Figure 7C). Subsequently, we examined whether Prox1 was able to inhibit the proliferation of breast cancer cells from this mouse model. In agreement with the data in MCF7 cells, Prox1 over-expression was not able to repress mmtv-Myc expression and, therefore, cannot inhibit cellular proliferation (Figure 7D–G). Thus, we concluded that the inhibitory action of Prox1 on c-Myc gene expression mediates the tumor-suppressing function of Prox1 in breast cancer cells.

### 3.8. Prox1 Suppresses Tumor Growth In Vivo

Subsequently, we wanted to test whether Prox1 was able to inhibit breast tumor growth in an orthotopic xenograft mouse model, which better mimics the original tumor microenvironment. To this end, we over-expressed GFP or Prox1 in MDA-MB-231 cells (adenoviral system) and we then performed surgical orthotopic inoculation in the left and right mammary glands, respectively, in 6-week-old female NOD/SCID mice (Figure 8A). The GFP over-expressing cells were sufficient to generate detectable tumors one week after the injection; so, we conducted measurements every four days after this time point. Consistently, the tumors generated from Prox1 over-expression were three times smaller than the GFP tumors (Figure 8B–D). They also exhibited significantly reduced levels of the proliferation markers Ki-67 and pH3 (Figure 8E–H). In agreement with our observation regarding the repressive action of Prox1 on c-Myc gene expression, the Prox1-tumors showed reduced levels of c-Myc immunostaining (Figure 8I,J). These observations suggest that Prox1 is able to impair the growth of breast tumor cells in vitro and in vivo, possibly by a direct inhibitory action on c-Myc gene expression.

To examine whether the ability of Prox1 to inhibit tumor growth depends on breast tissue microenvironment or it is a general property of Prox1, we employed a heterotopic xenograft mouse model. In this regard, we over-expressed GFP or Prox1 in MDA-MB-231 cancer cells (adenoviral system) and then transplanted them subcutaneously into NOD/SCID mice. After twenty days, tumors were detectable in the GFP over-expressing control condition, and therefore, tumor volume measurements were performed every four days. Importantly, we noticed that the Prox1-tumors were consistently two times smaller than the GFP ones (Appendix A). We confirmed these observations by measuring the tumor weight at the end of the experiment (Appendix A). We then quantified two mitotic markers (Ki-67 and pH3) after immunostaining the tumors and found that in Prox1 tumors, proliferation markers are significantly lower than in GFP tumors (Appendix A). Similarly, immunostainings of the tumors against antibodies for CD31 (Appendix A), VEGF (Appendix A), and OCCLUDIN (Appendix A) showed that Prox1 reduced all three markers, indicating a decrease in angiogenesis as well as in the EMT capacity of the breast cancer tumors.

Collectively, our observations suggest that Prox1 suppresses the c-Myc gene expression to inhibit the Warburg effect, proliferation, migration, and tumor growth in breast cancer cells (Figure 9).

## 4. Discussion

Oncogenic signaling reprograms cellular metabolism to support rapid proliferation rates, metastasis, and tumor growth [35,56]. Metabolic reprogramming towards high rates of glucose uptake and lactate production, even in the presence of sufficient levels of oxygen (known as the Warburg effect), is a critical hallmark of cancer cells [27,28,34,57]. Our study identified the transcription factor Prox1 as a negative regulator of the Warburg effect in breast cancer cells. Using different breast cancer cell lines, we showed that Prox1 reverses the Warburg effect by repressing glucose uptake and lactate secretion. Prox1 inhibits the Warburg effect by directly repressing the expression of the proto-oncogene c-Myc. Consistently, Prox1 is also sufficient to suppress the expression of a number of downstream targets of c-Myc that promote glycolysis and lactate production (Figure 5). Our observations suggest that Prox1 is a key transcription regulator for the prevention of cancer-specific metabolic reprogramming. Therefore, via this action, Prox1 may act as a tumor-inhibiting barrier in normal breast tissue and cells. This action may also explain the existence of a breast cancer susceptibility locus over the Prox1 gene regulatory elements in radiotherapy-treated children [25].

In agreement with such a role, it was previously reported that Prox1 expression is epigenetically repressed in breast cancer cells by DNA methylation [24]. We also showed here that Prox1 expression is reduced in breast tumors as compared to normal tissue and that Prox1 correlates with a favorable prognosis in patients. Most importantly, Prox1 over-expression reverses the malignant phenotype of breast cancer cells in heterotopic and orthotopic xenograft models by suppressing proliferation, migration, and tumor growth. These observations support a tumor-suppressing role of Prox1 in breast tissue. Consistently, *Prox1* knockdown in primary mammary stem/progenitor cells induced proliferation, clonogenic potential, and stemness [26]. Similarly, Prox1 exerted tumor-inhibiting actions in other tissues and organs, including the nervous system, pancreas, lung, and hematological cell lines [12,18,20,58]. Our data raise the intriguing possibility that the tumor-suppressing function of Prox1 in these tissues may be exerted by its effect on c-Myc and metabolism. Thus, it would be interesting to investigate whether Prox1 suppresses the Warburg effect and c-Myc gene expression in these cellular contexts. Notably, our previously published RNA-seq data for Prox1 over-expression in neuroblastoma cancer cells indicated that the glycolytic metabolic processes are one of the top GO categories for down-regulated genes [5].

To mechanistically explain the involvement of Prox1 in the regulatory network of tumor inhibitory factors, we initially examined its ability to control the expression of key players in cell proliferation and metabolism (Figure 5). Interestingly, we found that Prox1 specifically suppresses *c-Myc* gene expression and identified a number of genomic loci (Locus B, Locus D, and Locus F) over and near the c-Myc gene as sites of Prox1 recruitment. Most importantly, c-Myc over-expression is sufficient to rescue the anti-proliferative effect of Prox1. These data suggest a direct repressive action of Prox1 in c-Myc that inhibits tumorigenesis and metabolic reprogramming in breast cancer cells. It would also be interesting to investigate whether a cross-inhibitory relationship between c-Myc and Prox1 occurs. In particular, whether part of the oncogenic function of *c-Myc* is mediated by *Prox1* repression. Consistently, the *Drosophila* Myc induces proliferation and maintenance of neuroblasts by inhibiting Prospero’s (the *Drosophila* homolog of mammalian *Prox1*) activity. Genetic depletion of *Myc* in *Drosophila* neuroblasts promotes entry of Prospero into the nucleus and activation of their differentiation at the expense of proliferation [59]. Furthermore, based on our experimental observations, we cannot exclude additional regulatory actions of Prox1 on other genes or pathways that may contribute to its anti-tumorigenic role in breast cancer. Accordingly, in endothelial cells, Prox1 negatively regulates the metalloprotease MMP14, which is involved in cancer invasion and angiogenesis [60]. Thus, this action may participate in Prox1-mediated inhibition of metastasis in breast cancer. Nevertheless, we previously reported that Prox1 promotes the expression of the *CDKN1B* gene (encoding for p27-KIP1) in mouse and human neuroblastoma cells [12]. p27-KIP1 is a cyclin-dependent kinase inhibitor protein (CDKI) and a master negative regulator of cell cycle progression in mammalian cells [61]. However, in the context of breast cancer cells, Prox1 is not able to induce p27-KIP1 or other CDKIs, such as p21-CIP1 or p16. These data suggest that the cellular context dictates the ability of Prox1 to regulate downstream target genes and further support our conclusion that c-Myc is the major mediator of Prox1′s anti-proliferative effect on breast cancer cells. Additionally, Prox1 represses PDK1 and induces MPC1 in breast cancer cells. PDK1 promotes aerobic glycolysis as well as tumorigenesis and is a downstream target of c-Myc [48,49,62,63,64], whereas MPC1 inhibits the Warburg effect and exerts an important anti-tumorigenic role [51,65]. In future studies, it will be interesting to investigate whether Prox1 regulates the expression of these critical genes directly or indirectly, via its action on *c-Myc* gene expression.

Despite these anti-tumorigenic roles of Prox1, previous reports suggest that Prox1 promotes tumor initiation and progression in other cell types and tissues [11,23,66,67]. In particular, Prox1 expression has been correlated with tumorigenesis in colon, gastric, prostate, and liver cancers [17,23,68,69,70,71]. These findings highlight the complexity and context-dependent functions of Prox1 in tumor pathogenesis. We hypothesized that in these cell types, Prox1 is not able to interact with the regulatory elements of c-Myc and, therefore, to repress its expression. Moreover, the molecular mechanism by which a transcription factor could direct various or opposing regulatory effects on different cell types may be justified by the differential interaction of a large repertoire of protein partners that leads to differential control of downstream target genes. Consistent with this scenario, Prox1 has been reported to interact with many transcription factors, co-repressors, or co-activators in a cell-type specific manner, including LRH1 (NR5A2), SF1 (NR5A1), PAX6, COUPTFII, RORs, ERRa, PGC-1a, HNF-4a, PPARD, KLF2, HDAC3, LSD1, BRD3, BRD4, CHD4, CREBBP (CBP), and EP300 (p300) [72,73,74,75,76,77,78,79,80]. Consequently, Prox1 may orchestrate many distinct phenotypes, cellular processes, or signaling pathways depending on its expression pattern and that of its partners.

Conclusively, in this study, we revealed a key role for Prox1 in inhibiting breast cancer progression through the negative regulation of c-Myc. Moreover, we unraveled a previously unknown action of Prox1 in suppressing the Warburg effect on breast cancer cells that may also apply in other tissues, where Prox1 exhibits tumor-suppressive roles.

## 5. Conclusions

In this study, we uncovered a novel molecular mechanism that mediates tumor suppressive functions in breast tissue. In particular, we revealed a critical role for Prox1 in inhibiting breast cancer through the negative regulation of c-Myc at the transcriptional level. Through this action, Prox1 inhibited the proliferation and migration of breast cancer cells, as well as tumor growth in heterotopic and orthotopic xenograft models. Moreover, we unraveled a previously unknown action of Prox1 in suppressing the Warburg effect on breast cancer cells that may also apply in other tissues, where Prox1 exhibited tumor-suppressive roles.

## Figures and Tables

**Figure 1 cells-12-01869-f001:**
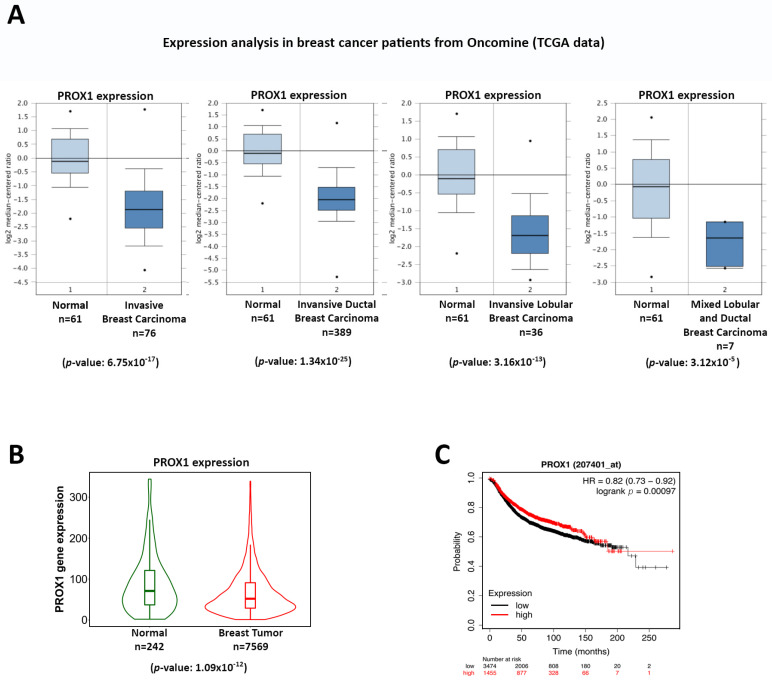
Prox1 expression is correlated with a favorable prognosis in breast cancer patients. (**A**) Expression of Prox1 mRNA in normal breast tissue and breast carcinoma. These data were downloaded from the Oncomine database and based on the TCGA data. *p* values for each comparison are indicated below the corresponding graph. (**B**) Graphical representation (violin plot) of Prox1 expression in breast tumors (n = 7569) and healthy adjacent breast tissue (n = 242) *p* < 0.001 (https://tnmplot.com, accessed on 1 October 2021). (**C**) Survival curve (Kaplan–Meier) of breast cancer patients with relative high and low expression of Prox1 from the KM-plotter (https://kmplot.com, accessed on 1 October 2021), *p* < 0.05.

**Figure 2 cells-12-01869-f002:**
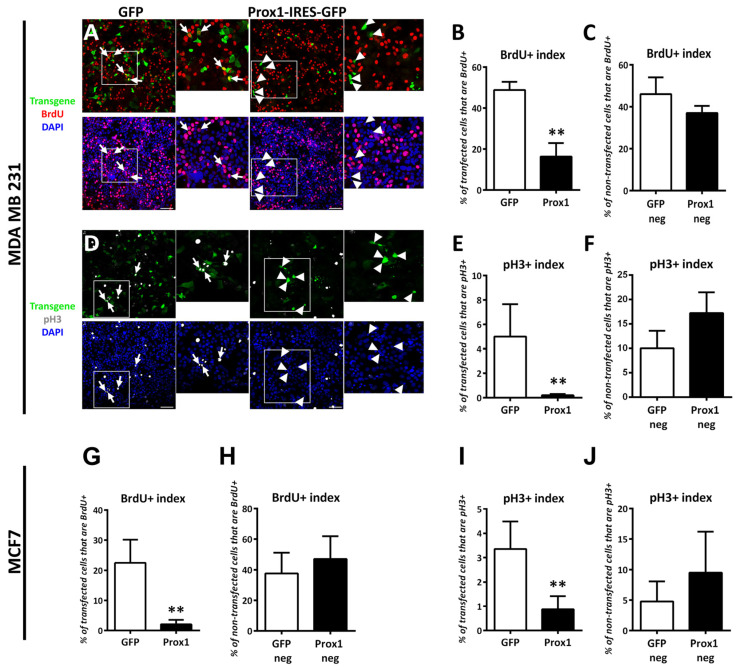
Prox1 inhibits proliferation of human breast cancer cells. (**A**) Prox1 and GFP-transfected MDA-MB-231 cells were treated with BrdU for 2 h and then stained with BrdU antibody (red) and labeled with DAPI (blue). Arrows indicate representative double positive cells (GFP positive and BrdU positive). Arrowheads indicate representative Prox1 transfected cells that are negative for BrdU. Larger magnifications of the areas included in the square shapes are presented in the micrographs next to each image. Scale bar: 75 μM. (**B**) Quantification of BrdU incorporation in transgene-positive MDA-MB-231 cells (GFP: 48.87255 ± 3.951479% vs. Prox1: 16.32653 ± 6.611298%, *p* < 0.01) (**C**) Quantification of BrdU incorporation in transgene-negative MDA-MB-231 cells (GFP: 46.07843 ± 8.008329% vs. Prox1: 37.04561 ± 3.412163%, *p* > 0.1). (**D**) Prox1 and GFP-transfected MDA-MB-231 cells were immunostained for pH3 (grey) and labeled with DAPI (blue). Arrows indicate representative double positive cells (GFP positive and pH3 positive). Arrowheads indicate representative Prox1 transfected cells that are negative for pH3. Larger magnifications of the areas included in the square shapes are presented in the micrographs next to each image. Scale bar: 75 μM. (**E**) Quantification of pH3-positive cells in transgene-positive MDA-MB-231 cells (GFP: 5 ± 2.659148% vs. Prox1: 0.2 ± 0.1%, *p* < 0.01). (**F**) Quantification of pH3-positive cells in transgene-negative MDA MB 231 cells (GFP: 10 ± 3.618435% vs. Prox1: 17.24138 ± 4.251835%, *p* > 0.1. (**G**) Quantification of BrdU incorporation in transgene-positive MCF7 cells (GFP: 22.51497 ± 7.616616% vs. Prox1: 2.066682 ± 1.49425%, *p* < 0.001). (**H**) Quantification of BrdU incorporation in transgene-negative MCF7 cells (GFP: 37.72455 ± 13.398910% vs. Prox1: 46.96707 ± 14.93805%, *p* > 0.1). (**I**) Quantification of pH3-positive cells in transgene-positive MCF7 cells (GFP: 3.353293 ± 1.13362% vs. Prox1: 0.8794391 ± 0.5358448%, *p* < 0.01). (**J**) Quantification of pH3-positive cells in transgene-negative MCF7 cells (GFP: 4.790419 ± 3.281816% vs. Prox1: 9.532062 ± 6.67863%, *p* > 0.1). For all cases, ** *p* < 0.01.

**Figure 3 cells-12-01869-f003:**
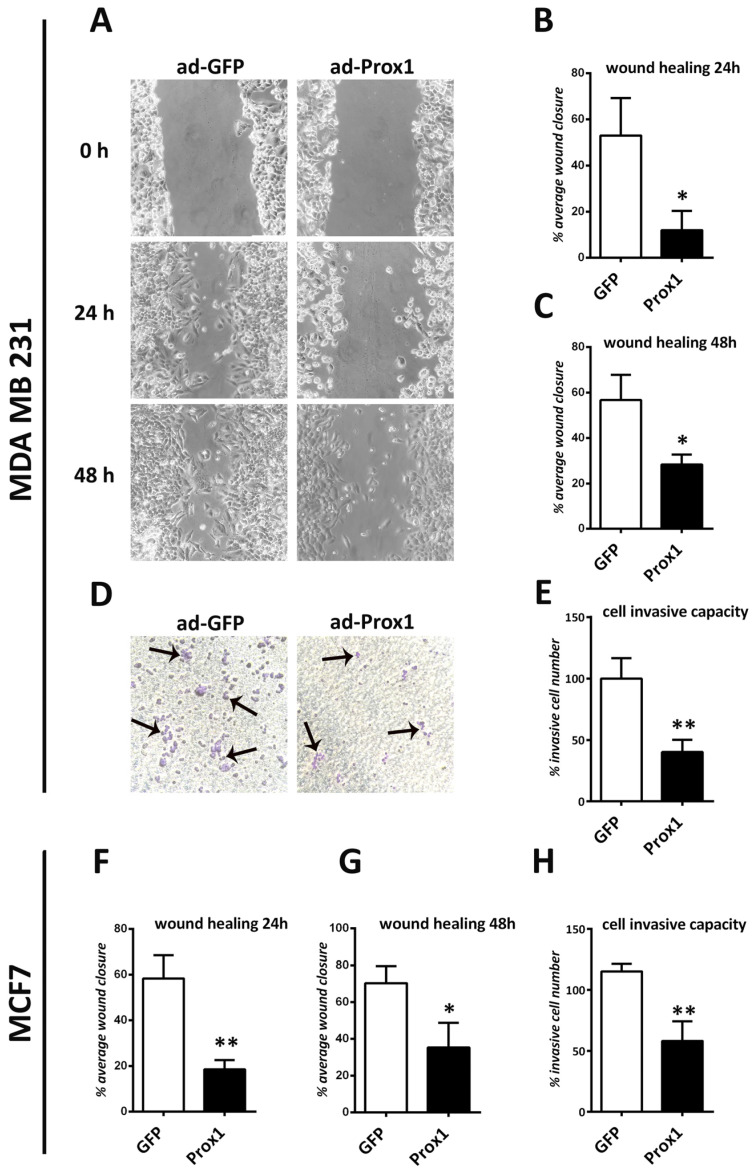
Prox1 suppresses the migration of human breast cancer cells. (**A**) Prox1 and GFP-infected MDA-MB-231 cells were measured for their migration capacity using the wound healing assay, at 24 and 48 h. (**B**) Quantification of the % average wound closure at 24 h in MDA-MB-231 cells in GFP and Prox1 over-expression conditions. (GFP: 52.99 ± 16.22% vs. Prox1: 12.02 ± 8.312%, *p* < 0.05). (**C**) Quantification of the % average wound closure at 48 h in MDA-MB-231 cells in GFP and Prox1 over-expression conditions (GFP: 56.76 ± 11.05% vs. Prox1: 28.31 ± 4.419%, *p* < 0.05). (**D**) Prox1 and GFP-infected MDA-MB-231 cells were measured for their cell invasion capacity using the transwell assay. Arrowheads indicate cells after invasion on the membrane of the transwell. (**E**) Quantification of the invasive cell number in MDA-MB-231 cells in GFP and Prox1 over-expression conditions (GFP: 100.0 ± 16.66% vs. Prox1: 40.11 ± 9.935%, *p* < 0.01). (**F**) Quantification of the % average wound closure at 24 h in MCF7 cells in GFP and Prox1 over-expression conditions (GFP: 58.39 ± 10.17% vs. Prox1: 18.62 ± 4.091%, *p* < 0.01) (**G**) Quantification of the % average wound closure at 48 h in MCF7 cells in GFP and Prox1 over-expression conditions (GFP: 70.29 ± 9.186% vs. Prox1: 35.39 ± 13.39%, *p* < 0.05) (**H**) Quantification of the invasive cell number in MCF7 cells in GFP and Prox1 over-expression conditions (GFP: 115.3 ± 6.242% vs. Prox1: 58.11 ± 16.14%, *p* < 0.01). For all cases, * *p* < 0.05, ** *p* < 0.01.

**Figure 4 cells-12-01869-f004:**
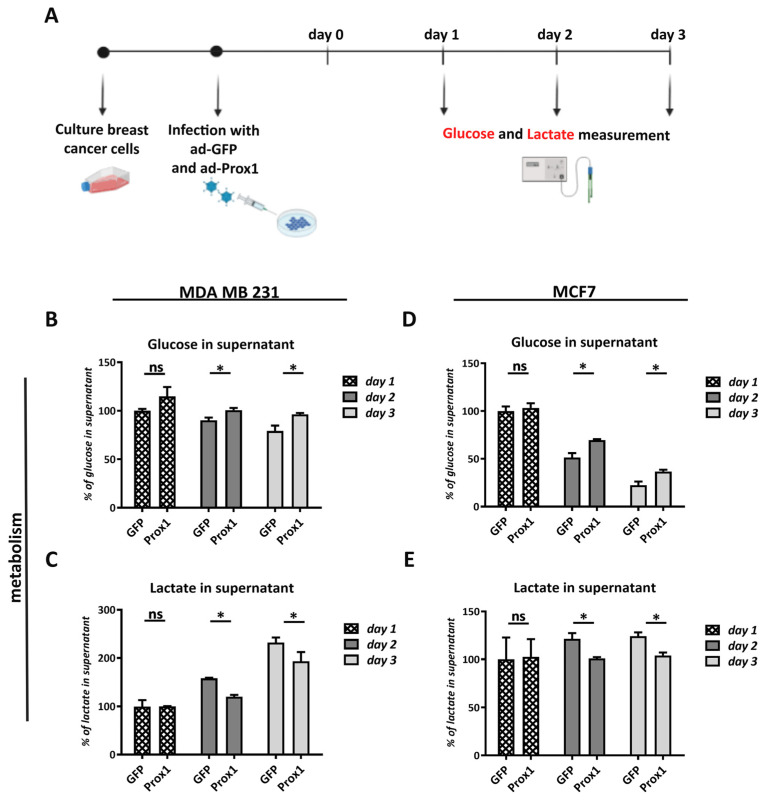
Prox1 over-expression reduces glucose and lactate production in the supernatant of human breast cancer cells. (**A**) Schematic representation of the experimental design, using the Biorender biotool. (**B**) Quantification of glucose in supernatant of MDA-MB-231 cells in GFP and Prox1 over-expression conditions. Measurements were conducted for three days in a row without changing the medium. The data are expressed as % of GFP condition in day 1 (GFP day1: 100.077 ± 1.97% vs. Prox1 day1: 114.801 ± 9.962%, *p* > 0.05; GFP day2: 90.184 ± 2.761% vs. Prox1 day2: 100.613 ± 2.316%, *p* < 0.05; GFP day3: 79.141 ± 5.598% vs. Prox1 day3: 96.319 ± 1.406%, *p* < 0.05). (**C**) Quantification of lactate in the supernatant of MDA-MB-231 cells in GFP and Prox1 over-expression conditions. Measurements were conducted for three days in a row without changing the medium. The data are expressed as % of GFP condition in day 1 (GFP day1: 99.712 ± 13.456% vs. Prox1 day1: 100 ± 0.511%, *p* > 0.05; GFP day2: 157.981 ± 0.982% vs. Prox1 day2: 120.123 ± 3.756%, *p* < 0.05; GFP day3: 232.244 ± 10.260% vs. Prox1 day3: 193.227 ± 19.270%, *p* < 0.05). (**D**) Quantification of glucose in the supernatant of MCF7 cells in GFP and Prox1 over-expression experimental conditions. Measurements were conducted for three days in a row without changing the medium. The data are expressed as % of GFP condition in day 1 (GFP day1: 100 ± 4.792% vs. Prox1 day1: 103.233 ± 4.923%, *p* > 0.05; GFP day2: 51.324 ± 4.636% vs. Prox1 day2: 69.592 ± 0.440%, *p* < 0.05; GFP day3: 22.374 ± 3.849% vs. Prox1 day3: 36.677 ± 1.995%, *p* < 0.05). (**E**) Quantification of lactate in the supernatant of MCF7 cells in GFP and Prox1 over-expression conditions. Measurements were conducted for three days in a row without changing the medium. The data are expressed as % of GFP condition in day 1 (GFP day1: 100 ± 22.803% vs. Prox1 day1: 102.566 ± 18.520%, *p* > 0.05; GFP day2: 121.372 ± 6.029% vs. Prox1 day2: 101.022 ± 1.233%, *p* < 0.05; GFP day3: 124.248 ± 3.908% vs. Prox1 day3: 104.120 ± 2.991%, *p* < 0.05). For all cases, ns (non-significant) for *p* > 0.05, * *p* < 0.05.

**Figure 5 cells-12-01869-f005:**
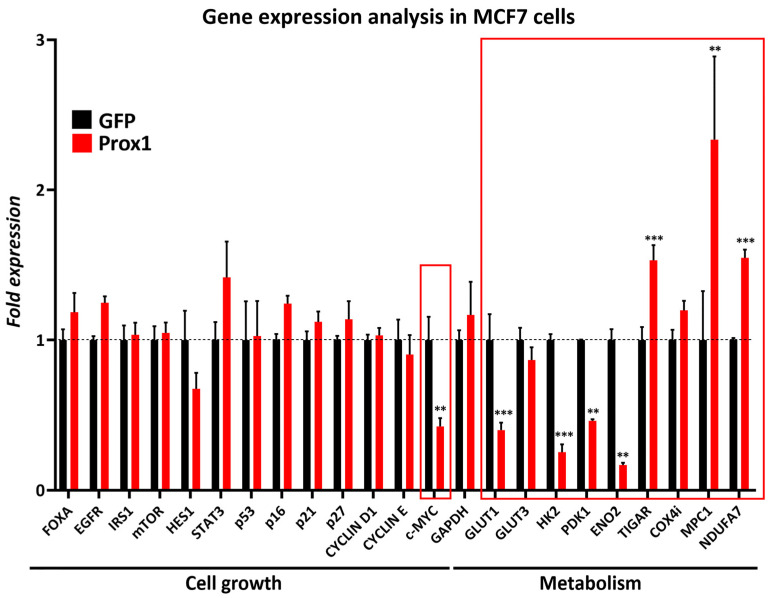
Prox1 inhibits expression of genes that promote the Warburg effect. mRNA expression analysis of genes critically involved in tumor development in MCF7 cells over-expressing GFP or Prox1. Relative expression levels of FOXA, EGFR, IRS1, mTOR, HES1, STAT3, p53, p16, p21, p27, CYCLIN D1, CYCLIN E, c-MYC, GAPDH, GLUT1, GLUT3, HK2, PDK1, ENO2, TIGAR, COX4i, MPC1, and NDUFA7 mRNA in GFP and Prox1 over-expression conditions, measured with quantitative real time RT-PCR. The red square highlights the genes that alter significantly their expression pattern. For all cases, ** *p* < 0.01, *** *p* < 0.001.

**Figure 6 cells-12-01869-f006:**
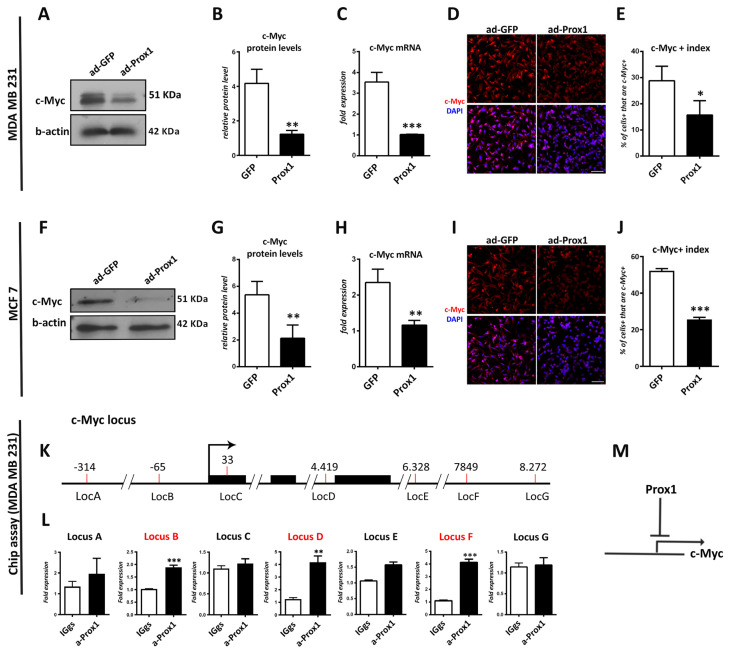
Prox1 directly suppresses c-Myc expression both in transcriptional and translational level. (**A**) Western blot analysis of c-Myc and b-actin in MDA-MB-231 cells in GFP and Prox1 over-expression conditions. (**B**) Quantification of protein expression levels of c-Myc in MDA-MB-231 cells in GFP and Prox1 over-expression conditions, *p* < 0.01. (**C**) Relative expression of c-Myc mRNA in MDA-MB-231 cells in GFP and Prox1 over-expression conditions, measured with quantitative real time RT-PCR, *p* < 0.001. (**D**) Prox1 and GFP-infected MDA-MB-231 cells were immunostained for c-Myc (red) and labeled with DAPI (blue). Scale bar: 75 μM. (**Ε**) Quantification of c-Myc positive cells over all cells in Prox1 and GFP-infected MDA-MB-231 cells (GFP: 28.78 ± 5.604% vs. Prox1: 15.71 ± 5.471%, *p* < 0.05). (**F**) Western blot analysis of c-Myc and b-actin in MCF7 cells in GFP and Prox1 over-expression conditions. (**G**) Quantification of protein expression levels of c-Myc in MCF7 cells in GFP and Prox1 over-expression conditions, *p* < 0.01. (**H**) Relative expression of c-Myc mRNA in MCF7 cells in GFP and Prox1 over-expression conditions, measured with quantitative real time RT-PCR, *p* < 0.01. (**I**) Prox1 and GFP-infected MCF7 cells were immunostained for c-Myc (red) and labeled with DAPI. Scale bar: 75 μM. (**J**) Quantification of c-Myc positive cells over all cells in Prox1 and GFP-infected MCF7 cells (GFP: 52.22 ± 1.217% vs. Prox1: 25.56 ± 1.217%, *p* < 0.05). For all cases, * *p* < 0.05, ** *p* < 0.01, *** *p* < 0.001 (**K**) Schematic representation of the c-Myc gene locus around the transcription start site (denoted with the broken arrow). The exons of the c-Myc gene are represented as black boxes. The genomic loci that we tested in ChIP experiments are shown with red lines. (**L**) ChIP analysis of the binding sites of Prox1 to c-Myc gene locus. ChIP experiments were performed using anti-Prox1 (a-Prox1) or a control antibody (IgG) in chromatin isolated from MDA-MB-231 cells over-expressing Prox1. For a-Prox1 and IgG reactions, the same amount of DNA was used as a template. The primers pairs used to amplify the corresponding DNA sequences are indicated with specific loci letters. Letters denote the distance from the transcription start site. Note that Prox1 specifically binds to the loci B, D and F (these loci are noted with red color). (**M**) Schematic representation of Prox1 directly suppressing c-Myc transcription.

**Figure 7 cells-12-01869-f007:**
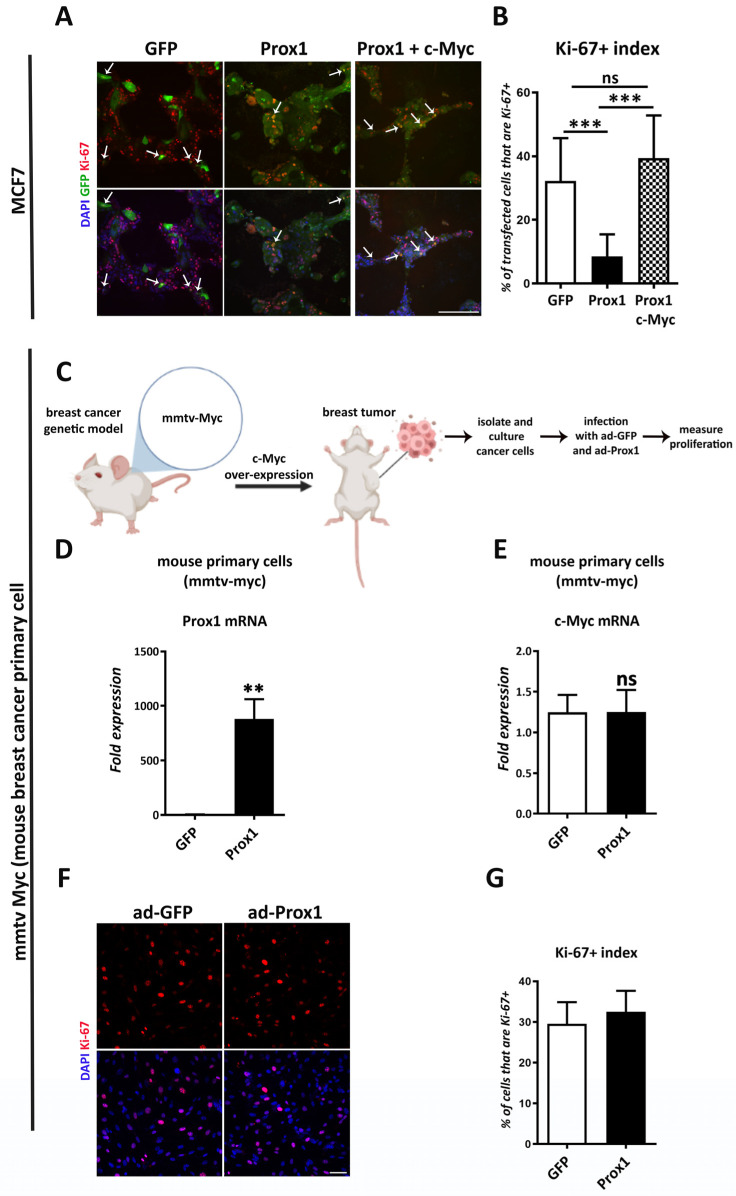
c-Myc over-expression rescues the Prox1-driven anti-proliferative effect of breast cancer cells. (**A**) Prox1, GFP, and Prox1 + c-Myc transfected MCF7 cells were immunostained for Ki-67 (red) and labeled with DAPI (blue). Arrowheads indicate representative double positive cells (GFP positive and Ki-67 positive or Prox1 and Ki-67 positive cells). Scale bar: 250 μM. (**B**) Quantification of Ki-67 positive cells in transgene-positive MCF7 cells (GFP% 32.18 ± 13.46% vs. Prox1: 8.511 ± 6.949%, *p* < 0.001; Prox1: 8.511 ± 6.949% vs. Prox1 + c-Myc: 42.78 ± 9.431%, *p* < 0.001). (**C**) Schematic representation of the in vivo breast cancer genetic mouse model and experimental design, using the Biorender biotool. (**D**) Relative expression levels of Prox1 mRNA in mouse breast cancer primary cells in GFP and Prox1 over-expression conditions, measured with quantitative real time RT-PCR, *p* < 0.01. (**E**) Relative expression levels of c-Myc mRNA in mouse breast cancer primary cells in GFP and Prox1 over-expression conditions, measured with quantitative real time RT-PCR, *p* > 0.1. (**F**) Prox1 and GFP-infected mouse breast cancer primary cells were immunostained for Ki-67 (red) and labeled with DAPI (blue). Scale bar: 75 μΜ. (**G**) Quantification of Ki-67 positive cells over all cells in mouse breast cancer primary cells in GFP and Prox1 over-expression conditions (GFP: 29.55 ± 5.333% vs. Prox1: 32.54 ± 5.157%, *p* > 0.1). For all cases, ns (non-significant) for *p* > 0.05, ** *p* < 0.01, *** *p* < 0.001.

**Figure 8 cells-12-01869-f008:**
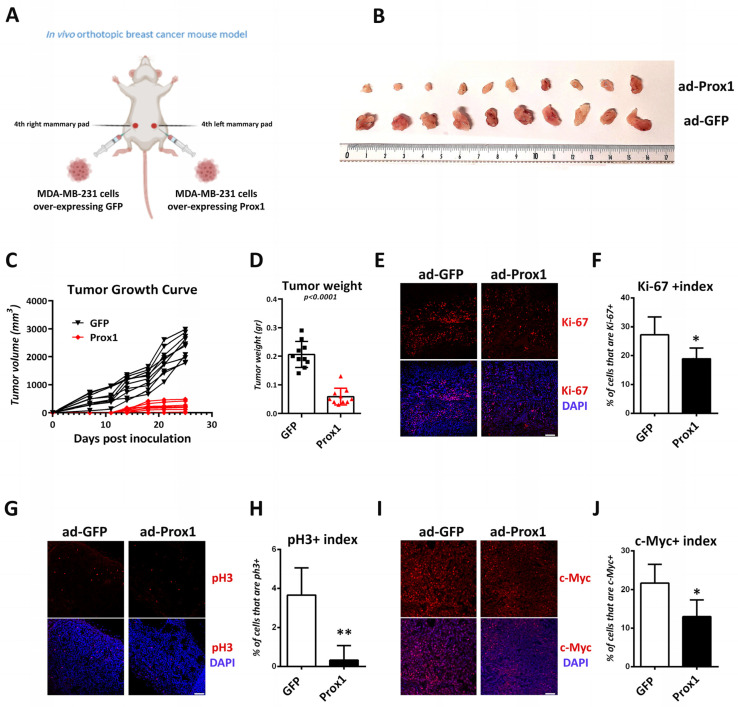
Prox1 inhibits proliferation of human breast cancer cells in vivo. (**A**) Schematic representation of the in vivo orthotopic breast cancer mouse model, using the Biorender biotool. (**B**) Representative images of whole tumors that were grown in NOD/SCID animals mammary pads using MDA-MB-231 cells over-expressing GFP or Prox1. (**C**) Quantification of the tumor growth after the orthotopic injections of MDA-MB-231 cells over-expressing GFP and Prox1, as indicated. (**D**) Quantification of the tumor weight of the tumors over-expressing GFP and Prox1 (GFP: 0.2060 ± 0.04526 g vs. Prox1: 0.05900 ± 0.02923 g, *p* < 0.0001). (**E**) Tumor sections of MDA-MB-231 cells over-expressing GFP and Prox1 were labeled for Ki-67 (red) and DAPI (blue). Tumors were collected at the end of the experiment. Scale bar: 40 μM. (**F**) Quantification of the Ki-67 index in GFP and Prox1 treated tumors (GFP: 27.22 ± 6.206% vs. Prox1: 18.89 ± 3.752%, *p* < 0.05). (**G**) Tumor sections of MDA-MB-231 cells over-expressing GFP and Prox1 were labeled for pH3 (red) and DAPI (blue). Tumors were collected at the end of the experiment. Scale bar: 40 μM. (**H**) Quantification of the pH3 index in GFP and Prox1 treated tumors (GFP: 3.667 ± 1.394% vs. Prox1: 0.3333 ± 0.7454%, *p* < 0.01). (**I**) Tumor sections of MDA-MB-231 cells over-expressing GFP and Prox1 were labeled for c-Myc (red) and DAPI (blue). Tumors were collected at the end of the experiment. Scale bar: 75 μM. (**J**) Quantification of the c-Myc index in GFP and Prox1-treated tumors (GFP: 21.67 ± 4.859% vs. Prox1: 13.00 ± 4.314%, *p* < 0.05). For all cases, * *p* < 0.05, ** *p* < 0.01.

**Figure 9 cells-12-01869-f009:**
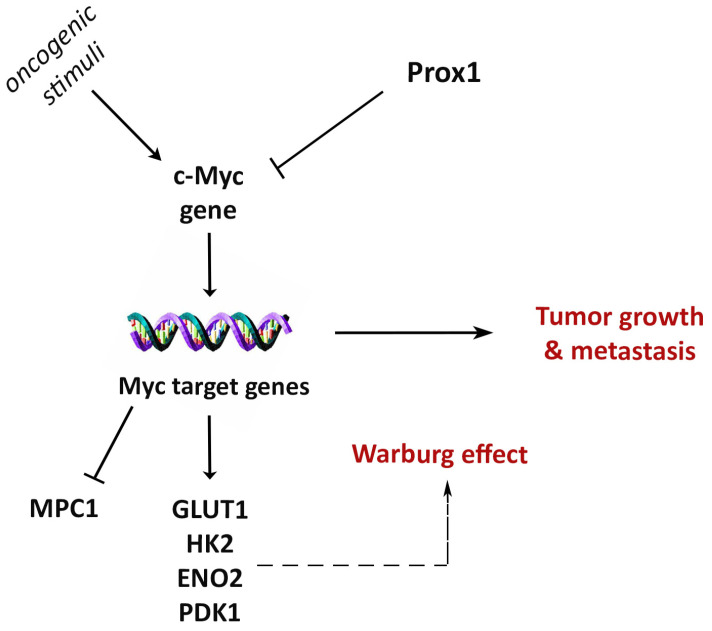
Proposed model of Prox1 anti-tumorigenic mechanism in breast cancer. Depiction of Prox1 inhibitory action on c-Myc and its effectors on Warburg effect. The negative effect of Prox1 on c-Myc gene expression represses tumor growth and metastasis.

**Table 1 cells-12-01869-t001:** Primers that were used for real-time RT-qPCR.

Genes	Sequence	
hPROX1	forward	ATCCCAGCTCCAATATGCTG
	reverse	TTGACGTGCGTACTTCTCCA
hCYCLIN D1	forward	CCCTCGGTGTCCTACTTCAA
	reverse	AGGAAGCGGTCCAGGTAGTT
hCYCLIN E1	forward	ATCCTCCAAAGTTGCACCAG
	reverse	AGGGGACTTAAACGCCACTT
hp27-kip1	forward	AGAGTTAACCCGGGACTTGG
	reverse	GCCCTCTAGGGGTTTGTGAT
hp21-cip1	forward	GGAAGACCATGTGGACCTGT
	reverse	GGCGTTTGGAGTGGTAGAAA
hIRS1	forward	ATGAGTGATGAGTTCCGCCC
	reverse	TGATGCTCTCAGTGCGTGAT
hEGFR	forward	ATGTGGTGACAGATCACGGC
	reverse	AGGCCCTTCGCACTTCTTAC
hFOXA	forward	TACGCAGACACGCAGGAG
	reverse	CCGCTCGTAGTCATGGTGTT
hmTOR	forward	AACGAGCTGGTCCGAATCAG
	reverse	AGGTTTTGTTCCGAAGCCCA
hc-MYC	forward	CGTCCTCGGATTCTCTGCTC
	reverse	GCCTGCCTCTTTTCCACAGA
hHES1	forward	ATGACAGTGAAGCACCTCCG
	reverse	CGTTCATGCACTCGCTGAAG
hGLUT-3	forward	TGGCCCAGATCTTTGGTCTG
	reverse	ATGGAAGGGCTGCACTTTGT
hGLUT-1	forward	TCTCGAAACTGGGCAAGTCC
	reverse	TCACCCACATACATGGGCAC
hHK2	forward	ACAAATTTCCGGGTCCTGCT
	reverse	TGTGGTCAAAGAGCTCGTCC
hPDK1	forward	TGCTGTATGGCCTGCAAGAT
	reverse	ACATTCTGGCTGGTGACAGG
hENO-2	forward	CCTGCAATGTGACGGAGAGT
	reverse	TTACACACGGCCAGAGACAC
hTigar	forward	GGTTGTAGAAGGCAAAGCGC
	reverse	ATTTTCACCTGGTCCAGCGT
hCOX4i	forward	GCAGCCTCTCCATGGATGAG
	reverse	CACAACCGTCTTCCACTCGT
hMPC1	forward	TCTTCCCATTGCTGCCATCA
	reverse	AACAGAAGCCAGTTCCGAGG
hNDUFA7	forward	TGCGACGACATGATGATGGA
	reverse	GCCCTCTAGGGGTTTGTGAT
hGAPDH2	forward	CCAGTATGAACTCCACTCACG
	reverse	CTCCTGGAAGATTGGTGATGG
hPPIA	forward	TGGACCCAACACAAATGGT
	reverse	ATGCCTTCTTTCACTTTGCC
hRPL13	forward	GCGGACCCGTGCCGAGGTTAT
	reverse	CACCATCCGCTTTTTCTTGCT
hp16	forward	GAGCAGCATGGAGCCTTC
	reverse	CATCATCATGACCTGGATCG
hSTAT3	forward	CAACTTCAGACCCGTCAACA
	reverse	CGATTCTCTCCTCCAGCATC

**Table 2 cells-12-01869-t002:** Quantitive PCR was used for detection and analysis of ChIP precipitates.

Primers Name	Orientation	Sequence
Locus A	forward	CCCCTCCCATATTCTCCCGT
	reverse	GAGACAAATCCCCTTTGCGC
Locus B	forward	CAGGCAGACACATCTCAGGG
	reverse	ACCTTCCACCCAGACTGAGT
Locus C	forward	ACTCAGTCTGGGTGGAAGGT
	reverse	CGTATACTTGGAGAGCGCGT
Locus D	forward	AGTGTTCTTGGTAAAGTCCCTCA
	reverse	GACATTGATGCCAATTCTTACCT
Locus E	forward	AGGCAGGTGAGAAGGTGAGA
	reverse	TCCTCACCTTCCTCCCAACT
Locus F	forward	TGACTCACTTGGGAATCGGG
	reverse	TGGAGAGTTGTTTTCCTTGCT
Locus G	forward	TGACTATGCCCTTTATCCATGACA
	reverse	TCCTTCCTCTTTATACATTCCATCCC

## Data Availability

The authors declare that all data supporting the findings of this study are available within the article and its Appendix A. All data presented in this study are available from the authors upon request.

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
