# Peer review of "Prox1 Suppresses the Proliferation of Breast Cancer Cells via Direct Inhibition of c-Myc Gene Expression"

_cells, 2023, doi:10.3390/cells12141869_

Round 1

Reviewer 1 Report

In the presented manuscript, the authors undertook to evaluate Prox1 as a negative regulator of breast cancer proliferation and metabolism. The authors showed that Prox1 inhibited breast cancer cell proliferation by directly inhibiting c-Myc gene expression. In my estimation, the experiments presented in the manuscript were well planned and conducted. A great advantage of the presented manuscript is the variety of experiments performed. The authors undertook to evaluate the presented research problem using in vitro and in vivo methods and statistical analysis of available databases. In my opinion, the obtained results are subject to appropriate statistical analysis and are clearly presented graphically in graphs/photos. The introduction to the role of Prox1 in the body, presented in the manuscript, adequately introduces the reader to the discussed topic and the undertaken research problem. In conclusion, I recommend this manuscript for publication in the journal Cells.

Author Response

We would like to thank the reviewer for the kind words, supportive remarks, and constructive comments. We truly appreciate the time and effort that the reviewer put into examining our work.

Reviewer 2 Report

In manuscript entitled „Prox1 Suppresses the Proliferation of Breast Cancer Cells via Direct Inhibition of c-Myc Gene Expression” the authors showed that Prox1 has a tumor suppressive role via direct transcriptional regulation of c-Myc, making it a promising therapeutic gene for breast cancer.

The topic is very interesting, and the authors showed some interesting results. The only one thing that is not clear is the methodology. Namely, the authors said that “The tumor in vivo model was established using 4-week-old both male and female NOD-SCID mice (NOD SCID gamma mouse)” – why in male mouse model? Also, there is not clear how many animals were divided and how many experimental groups were included in the study.

Generally, the manuscript is very well organized.

Author Response

We would like to thank the reviewer for the kind words, supportive remarks, and constructive comments. All of them have helped us improve our manuscript significantly. We truly appreciate the time and effort that the reviewer put into examining our work.

Regarding the comment about the use of female and male mice in the heterotopic xenograft experiments, we would like to thank the reviewer for pinpointing this mistake. We have used only female mice to study the effect of Prox1 overexpression in vivo. Thus, this statement in the methods section is a mistake that we have now corrected in the revised version of our manuscript. In particular, we have changed this description as follows (page 7, section 2.10.2): ”The tumor in vivo model was established using 4-week-old female NOD-SCID mice (NOD SCID gamma mouse) from the animal facilities of the Center for Experimental Surgery of the BRFAA.”.

Regarding the comment about the number of animals and experimental groups in the in vivo studies, we have now clarified this issue in the method section. In particular, we have added in the method section the following explanatory statements:

- (page 6, at the end of section 2.10.1): “For the in vivo orthotopic breast cancer mouse model, we used 10 female NOD-SCID mice (n=10).”

- (page 7, at the end of section 2.10.2): “For the in vivo heterotopic breast cancer mouse model, we used 5 NOD-SCID mice for each condition (GFP & Prox1), and the experimental protocol was performed for two times (n=20).”